# ITRAQ-Based Proteomic Analysis of Wheat (*Triticum aestivum*) Spikes in Response to *Tilletia controversa* Kühn and *Tilletia foetida* Kühn Infection, Causal Organisms of Dwarf Bunt and Common Bunt of Wheat

**DOI:** 10.3390/biology11060865

**Published:** 2022-06-05

**Authors:** Ting He, Tongshuo Xu, Ghulam Muhae-Ud-Din, Qingyun Guo, Taiguo Liu, Wanquan Chen, Li Gao

**Affiliations:** 1State Key Laboratory for Biology of Plant Disease and Insect Pests, Institute of Plant Protection, Chinese Academy of Agricultural Sciences, Beijing 100193, China; heting00329@163.com (T.H.); xuts1996@163.com (T.X.); gm3085pp@outlook.com (G.M.-U.-D.); tgliu@ippcaas.cn (T.L.); wqchen@ippcaas.cn (W.C.); 2Key Laboratory of Agricultural Integrated Pest Management, Qinghai University, Xining 810016, China; guoqingyunqh@163.com

**Keywords:** *Tilletia controversa*, *Tilletia foetida*, *Triticum aestivum*, proteomics, iTRAQ, dwarf bunt, common bunt

## Abstract

**Simple Summary:**

Wheat is the staple food crop for many countries. Therefore, avoiding pathogen infection in the crops is a prerequisite for productive and sustainable agriculture. Exploring the molecular mechanisms of pathogenesis related proteins and thaumatin-like proteins is significant for control of dwarf bunt and common bunt. In the present research, a highly susceptible cultivar was used to measure the defense protein expression differences against dwarf bunt and common bunt pathogens. There were huge differences in expression of defense proteins in pathogen-infected and control libraries.

**Abstract:**

Dwarf bunt and common bunt diseases of wheat are caused by *Tilletia controversa* Kühn and *Tilletia foetida* Kühn, respectively, and losses caused by these diseases can reach 70–80% in favourable conditions. *T. controversa* and *T. foetida* are fungal pathogens belonging to the *Exobasidiomycetes* within the basidiomycetous smut fungi (*Ustilaginomycotina*). In order to illuminate the proteomics differences of wheat spikes after the infection of *T. controversa* and *T. foetida*, the isobaric tags for relative and absolute quantification (iTRAQ) technique was used for better clarification. A total of 4553 proteins were differentially detected after *T. controversa* infection; 4100 were upregulated, and 453 were downregulated. After *T. foetida* infection, 804 differentially expressed proteins were detected; 447 were upregulated and 357 were downregulated. In-depth data analysis revealed that 44, 50 and 82 proteins after *T. controversa* and 9, 6 and 16 proteins after *T. foetida* were differentially expressed, which are antioxidant, plant-pathogen interaction and glutathione proteins, respectively, and 9 proteins showed results consistent with PRM. The top 20 KEGG enrichment pathways were identified after pathogen infection. On the basis of gene ontology, the upregulated proteins were linked with metabolic process, catalytic activity, transferase activity, photosynthetic membrane, extracellular region and oxidoreductase activity. The results expanded our understanding of the proteome in wheat spikes in response to *T. controversa* and *T. foetida* infection and provide a basis for further investigation for improving the defense mechanism of the wheat crops.

## 1. Introduction

Wheat is one of the most important staple food crops worldwide. In China, it has a healthy contribution in the economy of country [1]. Dwarf bunt of wheat (DBW) and common bunt of wheat (CBW) caused by *T. controversa* and *T. foetida*, respectively, are widespread in China and teliospores of both fungus have ability to survive for 3–10 years in soil under favourable conditions [2,3]. The occurrence of DBW is location sporadic and specific, while CBW is located throughout wheat growing areas. Yield losses caused by these diseases can reach 70–80% or total loss of wheat production in a severe attack [4,5]. DBW and CBW cause leaf flecking, extreme dwarfing, increased tillering and reduced quality and quantity of kernels by releasing bunt ball teliospores [6,7,8] and are mostly noted in cold wheat growing regions of world [9]. Both diseases have been reported to be as high in China, USA and other wheat growing areas [2,10]. DBW and CBW are now an increasing threat to wheat cultivation in America, Canada, Europe and Asia [11]. *T. controversa* and *T. foetida* absorbed nutrients from cells and tissues for establishment and colonisation and can release many effector proteins into host cells that inhibit plant defense responses, and these effector proteins may be involved in cellular functions where they perform different functions [12,13]. For example, host transcription, immune responses and chromatin remodelling may be affected by secreted effectors [14]. 

The pathogenic fungi can release a huge number of proteins, with only a few proteins characterised as the effector proteins. *Magnaporthe oryzae* was the first fungus in which several effector proteins were reported, including MoHEG13, Slp1 and MoHEG16, which are involved in the regulation of secondary metabolism of their hosts [15,16]. In the smut fungi, such as *Ustilago maydis*, many effector proteins have been reported, including See1, Cmu1, Pit2 and Tin2. These effector proteins are taken up by the plant cells, and have ability to spread into neighbouring cells and alter the metabolic process of these neighbouring cells through metabolic priming [17,18,19,20,21]. Pit2 effector protein has a role in plant immune responses by controlling cellular proteases of the host plants and is essential for maintenance of biotrophy and induction of tumours. Pit2 functions as an inhibitor of proteases, whose action is directly related with salicylic acid-associated plant defenses [17]. See1 fungal effector contributes to the formation of maize leaf tumours [18]. Similarly, Pep1 fungal effector protein has a role in inhibition of host peroxidase activity and suppresses the plant immunity [19]. However, no information is known about the mechanisms of action of *T. controversa* and *T. foetida* effector proteins. 

The plant receptor proteins can actively recognise pathogen effector proteins, and functions of these plant receptor proteins have been reported in many studies. For instance, plant receptor proteins Cf-2, Cf-4, Cf-9 and Cf-4E interact with pathogen effector proteins Avr2, Avr4, Avr9 and Avr4E, respectively, in tomato leaf mould disease caused by *Cladosporium fulvum* [22,23]. Eleven effector proteins have been secreted by *Ustilaginoidea virens* to induce non-host cell death in *Nicotiana benthamiana* [24]. Additionally, effector proteins are mostly recognised in plant defence signalling pathways, including salicylic acid, jasmonic acid and ethylene [25]. *Tilletia horrida* encodes 131 out of 597 proteins as an effector protein [26]. Additionally, many effector genes have roles in the establishment of successful infection of *T. horrida* [27]. However, according to our knowledge, no effector genes of *T. controversa* and *T. foetida* have been functionally characterised until now. The iTRAQ technology has been widely used in differential proteomic analysis of plant–pathogen interaction. In the interaction of pepper and *Bemisia tabaci,* several DEPs were expressed, including protein metabolism, carbon metabolism, redox regulation, stress response and carbon metabolism [28]. In *Arabidopsis* and *Verticillium dahlia*, 780 differentially accumulated proteins were expressed [29]. The primary aim of this study was to determine how the proteomics structure of the wheat crops changes after *T. controversa* and *T. foetida* infection. In this study, proteomics changes were observed in wheat spikes after *T. controversa* and *T. foetida* using iTRAQ technology. 

## 2. Materials and Methods

### 2.1. Spore Preparation

The pathogenic fungi *T. controversa* and *T. foetida* were cultivated in an incubator for spore suspension to inoculate wheat plants by following the method of our previous studies [9,30]. Briefly, *T. controversa* were grown on 2% soil media at 5 °C for 60 days under 24 h light regime and *T. foetida* were grown on 2% agar media at 15 °C for 15 days under 24 h light regime in an incubator (MLR 352 H, Panasonic, Osaka, Japan). The spore germination of both fungi were visualised under an automated inverted fluorescence microscope (IX83, Olympus, Tokyo, Japan), which were induced at their respective media. The spores of both fungi were harvested in laminar flow by adding 5 mL of ddH_2_O in every culture plate of *T. controversa* and *T. foetida*.

### 2.2. Plant Material and Spore Inoculation

Wheat cv. Dongxuan 3 (highly susceptible) grains were collected from the seed bank of the Institute of Plant Protection, Chinese Academy of Agricultural Sciences, Beijing-China. The Dongxuan 3 grains were surface sterilised with 30% sodium hypochlorite (NaClO) for 5 min and washed with double distilled water (ddH_2_O) thrice. The 30 grains were grown in petri plates, and plates were wrapped with aluminium foil and placed in an incubator to vernalise. The vernalisation was performed under 24 h light and 60% relative humidity at 4 °C for 30 days. After vernalisation, seedlings were transplanted into pots filled with sandy loam soil and organic matter at a ratio of 1:2% and grown in growth chambers (ARC-36, Percival, IA, USA). Twelve seedlings were transplanted into every pot. Ten pots were used for *T. controversa*, ten pots were used for *T. foetida* and ten pots were used as a control. Wheat seedlings were grown in incubator (ARC-36, Percival, IA, USA) at a 24 h light cycle at 8 °C at the seedling stage, 15 °C at the jointing stage and 20 °C at the booting stage. The fungal spores of both pathogens at a concentration of 10^6^ spores/mL with an OD_600_ of 0.15 was used to inject the wheat kernels for consecutive 5 days, while wheat spikes injected with ddH_2_O were used as a control in this study. Ten wheat kernels of both *T. controversa*- and *T. foetida*-infected plants were collected for further processing. Three biological replicates and three technical replicates were used for fungal-infected and control samples. 

### 2.3. Molecular Detection of T. controversa and T. foetida in Wheat Spikes Using Conventional PCR

DNA was extracted from wheat spikes at the flowering stage (Z65) using the Plant Genomic DNA Extraction Kit (TianGen, Beijing, China). The quality and concentration of extracted DNA were checked based on the 260/230 nm and 260/280 nm absorbance ratios using NanoDrop 2000 machine (Thermo Scientific, USA). PCR was done in triplicate with a total volume of 25 µL, including 1 µL of DNA (100 ng/µL), 1 µL of forward primer (10 µM), 1 µL of reverse primer (10 µM), 12.5 µL of master mix (TransGen, Beijing, China) and 9.5 µL of ddH_2_O (TransGen, Beijing, China). The PCR cycling conditions were followed as described by Ren et al., 2021 [31]. The primer sequences for *T. controversa* were ISSR859-140AF-5′TGGTGGTCGGGAAAGATTAGA-3′ and ISSR859-511AR- 5′-GGGACGAAGGCATCAAGAAG-3′ [32], and for *T. foetida* were L60F (5′-TCACTTCAAGGTCGTTCCCG-3′)/L60R (5-CGGGTCGAGGGGCGTAAACTTGA-3′) [33]. The PCR product (10 µL) was mixed with the help of pipette with 2 µL of 6 × Loading Buffer (TransGen, Beijing, China) and ran on a 1.5% agarose gel for electrophoresis analysis. 

### 2.4. Protein Extraction and Sample Preparation

Five spikelets were dipped in liquid nitrogen and grinded, and protease inhibitor and extraction buffer were added in a ratio of 0.5 mol/L Tris–HCl pH 8.3 and vortexed for 10 min at 4 °C. The same volume of Tris-saturated phenol (pH 8.0) was added, vortexed for 12 min through vortex machine (IKA ^®^ VORTEX 3, Beijing-China), centrifuged (Beijing, China) at 15,000× *g* for 20 min at 4 °C for phenol phase, transfer phenol phase was transferred into a new sterilised centrifuge tube, a certain amount of extraction buffer was added, vortexed for 12 min through vortex machine (IKA ^®^ VORTEX 3, Beijing-China), centrifuged (Beijing, China) at 15,000× *g* for 20 min at 4 °C for phenol phase, phenol phase was transferred into a new sterilised centrifuge tube, pre-chilled ammonium acetate methanol solution was added in a certain amount and protein was precipitated overnight at −20 °C and centrifuged at 15,000× *g* at 4 °C for 20 min. Supernatant was discarded, and pre-chilled (−20 °C) 90% acetone was added and vortexed to mix and washed thrice. The pellet was suspended with a certain amount of lysate (protease inhibitor and 8 M urea + 1% SDS) to completely dissolve the sample protein and centrifuged at 15,000× *g* at 4 °C for 20 min, and the supernatant was collected. 

The lysis buffer (1 × protease inhibitor, 8 M urea and 1% SDS) was added into the samples. The lysis procedure was done through sonication on ice for 3 min and kept on ice for 30 min. The collected supernatant was transferred into a new Eppendorf tube after centrifugation at 15,000× *g* at 4 °C for 15 min. 

### 2.5. The Protein Digestion and iTRAQ Labelling

As the schematic diagram described (Appendix A), the bicinchoninic acid (BCA) protein method was used to evaluate protein concentration, and then 100 µg protein was shifted to a sterilised new Eppendorf tube to maintain a total volume of 90 µL in the presence of 8 M urea and 1% SDS and vortexed for 30 s, and 10 µL of 1 M TEAB solution was added to the final volume by maintaining the pH 8 of each sample. Two microliters of 0.5 M TCEP (10 mM) was added to each sample, centrifuged shortly through thermomixer, incubated for 1 h at 37 °C, 1 M iodoacetamide (4 μL) solution was added to each sample and samples were kept at room temperature for 40 min. The pre-cooled acetone was added in the ratio of 1:6 and gently mixed to precipitate the proteins at 20 °C for 4 h. Pre-chilled 90% acetone aqueous solution was used to wash precipitate twice, and 100 µL of 100 mM triethylammonium bicarbonate (TEAB) solution was used to resolve. The protein was digested at 37 °C overnight by adding sequence grade modified trypsin (Promega, Madison, WI, USA) in the ratio of 1:50 (enzyme:protein, weight:weight). The resultant peptide mixtures were labelled with an iTRAQ 8Plex labelling kit (Sciex, Framingham, MA, USA) following the manufacturer’s guidelines. The labelled peptide samples were then pooled and lyophilised in a vacuum concentrator.

### 2.6. High pH Reversed Phase Separation

The labelled peptide samples were placed in a vacuum concentrator and lyophilized. The peptide samples were dissolved in buffer A (buffer A: 20 mM ammonium formate aqueous solution, adjusted to pH 10.0 with ammonia), and the reverse column (XBridge C18 column, 4.6 mm × 250) was connected with an Ultimate 3000 system (ThermoFisher scientific, Waltham, MA, USA). For high pH separation, a linear gradient, 5% B to 45% B within 40 min (B: 80% ACN added in 20 mM ammonium formate, ammonia hydroxide adjusted to pH 10.0) was used. The column was initially treated for 15 min, the column flow rate was adjusted to 1 mL/min and the temperature of column was adjusted to 30 °C. After 40 fractions were collected, they were cross-combined into 12 fractions, and each fraction was dried in a vacuum concentrator for the next step. 

### 2.7. Ultra-High-Performance Liquid Chromatography (UHPLC)-MS/MS Analysis

The peptides were re-dissolved in solvent A, which contained 0.1% formic acid, and line nanospray LC-MS/MS was used for analysis on the Q Exactive ^TM^ (Waters, Milford, MA, USA) coupled to the UHPLC BEH Amide system (Waters, Milford, MA, USA). The peptide in the quantity of 2.5 µL was loaded, and a 90-min gradient was used to separate the peptides from 5 to 35% B (B: 0.1% formic acid in 100% CAN). The flow rate of the column was adjusted at 300 nL/min for running. The mass spectrometer (MS/M) was used with the power of 2 kV electrospray voltage. The MS/M was run in acquisition mode and adjusted to switch automatically between MS and MS/MS modes. The parameters were: (1) MS: resolution= 70,000; scan range (m/z) = 300–1800; maximum injection time = 60 ms; AGC target = 3 × 10^6^ dynamic exclusion = 20 s; include charge states = 2–7; (2) HCD-MS/MS: isolation window = 2.2; resolution = 17,500; AGC target = 5 × 10^4^; collision energy = 30 and maximum injection time = 80 ms.

### 2.8. Proteomic Data Analysis

The software PEAKS Studio (Bioinformatics Solution Inc., Waterloo, Canada) version 8.5 was used for tandem mass spectra for proteomic analysis. PEAKS DB was established to examine uniprotproteome_UP000009096 databases (17,877 entries) presumptuous trypsin as the digestion system [34]. PEAKS DB were searched with a fragrant ion mass tolerance of 0.05 Da and a parent ion tolerance of 10 ppm. Carbamdomethylation (C) and Itraq 8plex (K, N-term) were specified as the fixed modifications. Oxidation (M) and deamidation (NQ) were specified as the variable modifications. The steps of PEAKS DB were followed as previously reported [35], while 1% FDR and 1 unique was used for filtering the peptides for normalising the data. ANOVA was used for protein and peptide abundance calculation, *p* ≤ 0.05. Normalisation was achieved on averaging the abundance of all peptides. Medians were used for averaging. Proteins with fold change in a comparison >1.2 or <0.83 and adjusted significance level *p* < 0.05 were considered deferentially expressed [36]. The raw data were available on ProteomeXchange https://www.iprox.cn/cas/login?service=https%3A%2F%2Fwww.iprox.cn%2Flogin%2Fcas (accessed on 19 May 2022, Username: Sara, code: SARA1880016, Project ID IPX0004463000).

### 2.9. Protein Function Annotation

The GO term and KEGG pathway analysis were followed as previously reported, with franklin delano roosevelt (FDR) < 0.05 representing the significantly expressed genes [10,37,38]. 

### 2.10. Parallel Reaction Monitoring (PRM) Verification Analysis

Nine proteins, including 1 reference protein, were selected for validation of proteomics results by PRM on Triple TOF 5600+ LC-MS/MS system (SCIEX). PRM analysis was performed in the same sequence as in the iTRAQ experiment using peptide quantification kit (Thermo Fisher Scientific, MA, USA). The peptides were dissolved in solvent A (A: 0.1% formic acid aqueous solution) after the salt was lyophilised and analysed by LC-MS/MS equipped with an online nano jet ion source. The size of the sample was 2 µL and separated under gradient condition of 120 min. The samples were run at 40 °C with a column flow rate of 400 nL/min in an electrospray voltage of 2 kV. Spectro Dive 10.0 was used for PRM data analysis. The credibility card value for peptide identification was Q ≤ 0.01. The protein expression level was normalised by the total TIC of each sample extracted by PEAKS Studio. 

## 3. Results

### 3.1. Pathogen Identification

The spikes at the flowering stage (Z65) were collected to extract DNA for PCR, and a specific single band of *T. controversa* 372 bp and *T. foetida* 66 bp were detected (Figure 1) in infected samples, indicating that the infection of both pathogens was successful. The image of wheat spikes is shown in Appendix A.

### 3.2. Identification of Significantly Differently Expressed Proteins (DEPs) on iTRAQ Technology

To identify DEPs of wheat against *T. controversa* and *T. foetida*, the quantitative and qualitative proteomic analysis carried out using iTRAQ/tandem mass tag technology. After *T. controversa* infection, a total of 4553 DEPs were obtained, of which 4100 were upregulated and 453 were downregulated (Figure 2a, Appendix A), and after *T. foetida* infection, a total of 804 DEPs were obtained, of which 447 were upregulated and 375 were downregulated (Figure 2b, Appendix A) at 1 Unique Peptide and Peptide Threshold 1.0% FDR. 

### 3.3. GO Enrichment Analysis of DEPs

GO functional annotation was characterised into three groups: biological process (BP), cellular component (CC) and molecular function (MF). The summarised GO mapping and annotation data of differentially expressed proteins at GO level after *T. controversa* and *T. foetida* are shown in Figure 3, and the smallest Q values were included in the 30 top GO graph. For *T. controversa*, DEPs in the biological process were highly enriched in metabolic process, cellular process and single organism process, localisation and biological regulation. In the cellular component, the enriched DEPs were cell, cell part, organelle and macromolecular complex. Similarly, in the molecular function, DEPs were mainly distributed in catalytic activity, binding, transporter activity and structural molecule activity (Figure 3a). For *T. foetida*, DEPs in the biological process were highly enriched in the carbohydrate metabolic process, oxylipin biosynthetic process, starch biosynthetic process, lipid oxidation and glycogen biosynthesis process. In the cellular component, DEPs were mostly related to the extracellular region, peroxisome and lysosome. Similarly, under the category of molecular function, DEPs were mostly related to oxidoreductase activity, serine-type endopeptidase inhibitor activity, pyridoxal phosphate binding, oxidoreductase activity, acting on the aldehyde or oxo group of donors, disulfide as acceptor and NAD(P)H dehydrogenase (quinone) activity (Figure 3b). 

### 3.4. KEGG Pathway Enrichment Analysis of DEPs

To identify the biological pathways after *T. controversa* and *T. foetida* infection, these proteins were further mapped to the corresponding pathways included in the KEGG database. For *T. controversa*, the top 20 KEGG enrichment pathways were metabolic pathways, carbon metabolism, amino sugar and nucleotide sugar metabolism, and cysteine and methionine metabolism were expressed highly (Figure 4a). For *T. foetida*, the top 20 KEGG enrichment pathways were starch and sucrose metabolism (ats00592), glycolysis/gluconeogenesis (ats00010), valine, leucine and isoleucine degradation (ats00280) and pyruvate metabolism (ats00620), and alpha-linolenic acid metabolism (ats00592) were significantly enriched pathways (Figure 4b). 

### 3.5. Differentially Expressed Proteins (DEPs) in Response to T. controversa and T. foetida Infection

Following *T. controversa* and *T. foetida* infection, we identified significant DEPs in *T. controversa*- and *T. foetida*-infected and control libraries. Most DEPs were downregulated after *T. controversa* infection. Results showed that 36 out of 43 and 7 out of 43 antioxidant proteins, 47 out of 50 and 3 out of 50 plant-pathogen interaction proteins and 75 out of 82 and 7 out of 82 glutathione metabolism proteins were upregulated and downregulated, respectively, after *T. controversa* infection. Similarly, 5 out of 10 and 5 out of 10 heat shock proteins were upregulated and downregulated, respectively (Appendix A). For *T. foetida*, most DEPs were upregulated after infection. Results showed that 2 out of 9 and 7 out of 9 antioxidant proteins, 1 out of 6 and 5 out of 6 plant-pathogen interaction proteins and 1 out of 16 and 15 out of 16 glutathione metabolism proteins were down-regulated and upregulated, respectively after *T. foetida* infection (Appendix A).

### 3.6. PRM Analysis to Verify DEPs

The results of iTRAQ analysis were confirmed by using the PRM techniques in the experiment. Based on the results of iTRAQ analysis, 9 proteins were upregulated and 1 protein was downregulated. The results of 9 proteins of PRM were consistent with the results of iTRAQ; however, quantification differences were noted in the expression levels of both iTRAQ and PRM, but trends was similar. However, protein (TraesCS2D02G123300.1) was upregulated during iTRAQ analysis, but was downregulated in PRM analysis (Table 1). 

## 4. Discussion

The fungal effector proteins have keys roles in the interaction mechanism of host-pathogen [39]. The sowing of resistant and tolerant plants against fungal pathogens is the best strategy to control diseases [40]. Until now, methods to determine the defense mechanisms of plants to *T. controversa* and *T. foetida* have been very limited, particularly those of susceptible plants. The proteomics studies have provided a new and strong tool to determine physiological alterations at the cellular level, but limited efforts have been made to recognise the response of wheat to *T. controversa* and *T. foetida* infection at the proteome level. Yin et al., [41] showed that 20 proteins were altered in *A. thaliana* after *B. tabaci* infection. To the best of our knowledge, it is the first time that iTRAQ was used to study the proteomics alterations in wheat spikes after *T. controversa* and *T. foetida* infection. 

Reactive oxygen species (ROS) have a relatively important role in maintaining the normal plant development and growth and increase the resistance against various stresses by activation of antioxidant proteins [42], and its abundance increased in resistance and susceptible plants after pathogen infection [43]. When plants grow in stressful conditions, ROS are often produced as metabolic byproducts [44]. When electrons from the electron transport chain in mitochondria and chloroplasts leak and react with the O_2_ molecule without other electron acceptors, ROS, such as superoxide, hydrogen peroxide, hydroxyl and singlet oxygen are generated, which activates the defence system of plants [45]. Accumulation of ROS causes oxidative damage in plants (nucleic acids, proteins and lipids) and causes degradation of chlorophyll pigments [46]. Therefore, the generation of ROS files must remain within the limits of factory compatibility. ROS reaches toxic levels under all forms of biotic stress and can cause explosions of antioxidants in plant cells. ROS activates the defense system of plants through the activation of plant phytohormones, ethylene, ascorbic acid and cytokinins. A complex network of defence and repair mechanisms counter these oxidation reactions [47]; however, any difference between ROS production and safe detoxification indicates a metabolic state called oxidative stress, which causes oxidative damage to the plant. In this study, we found that 32 and 8 peroxidase enzymes were upregulated after *T. controversa* and *T. foetida*, respectively, indicating that the actively participate in the plant’s defence response. Similarly, superoxide dismutase was upregulated after both *T. controversa* and *T. foetida* infections, which are known as first line of defence against oxidative stresses in plants and have key roles in hunting the ROS produced during biotic and abiotic stresses [48]. Additionally, peroxidase and catalase proteins were also identified in iTRAQ analysis of immune responses to plant virus infection [49]. The catalase enzymes have roles in the breakdown of hydrogen peroxide (H_2_O_2_) to oxygen (O_2_) and water (H_2_O) molecules [50]. The expression of catalase 2 (*Cat2*) gene in transgenic tobacco plants was higher compared to non-transgenic tobacco plants [51]. Additionally, the catalase enzymes have antioxidant action and can prevent the formation of free radicals, sequester them or even degrade them, thus minimising or preventing damage to plant cells [52]. The monodehydroascorbate reductase enzymes have important roles in plant resistance through breakdown of H_2_O_2_ to H_2_O and O_2_ molecules [53,54]. A similar response was noted in our results, in which expression of catalase and monodehydroascorbate reductase was higher in *T. controversa*- and *T. foetida*-infected plants (Appendix A). The plant pathogen interaction proteins (PRIP) recognise the elicitor molecules and activate the defence mechanism of plants [55]. Mitogen-activated protein kinases (MAPK) recognise the AvrRpt2 type III effector protein from *Pseudomonas syringae*, the causal organism of bacterial canker, and also indirectly interact with other nontoxic proteins [56]. The disease resistance proteins (DRP) robust defence mechanism against fungal pathogens by activation of signalling responses is known as Triggered Immune (PTI) Pathogen-Associated Molecular Pattern (PAMP) response, which eventually leads to changes in the gene expression of the host [57]. Pathogenesis-related proteins (PRs) have key roles in the defence mechanism of the wheat crops against *T. controversa*. PRs are a group of functionally diverse inducible proteins that accumulate in plant tissue in response to fungal infection [58]. The expression profiles of PRs was significantly higher in *T. controversa* infected plants than control plants [9]. Additionally, glutathione metabolism is an antioxidant molecule that plays a vital role in eliminating the ROS and harmful exogenous substances [59], involved in resistance against *Fusarium oxysporum* f. sp. *vasinfectum* [60] and drought stress [61]. Similarly, glutathione play key roles within different cell compartments, in the development of resistance and activating the plant defence. The accumulation of glutathione in peroxisomes and chloroplasts at the early stages of plant pathogen interactions is related to increased resistance and tolerance against different pathogens [62]. In this study, the expression of PRIP, MAPK, DRP and glutathione were changed by *T. controversa* and *T. foetida* infection, and these proteins may have roles in disease suppression caused by these pathogens. The heat shock cognate proteins (HSP) are highly conserved and widely distributed functional proteins induced under adverse conditions. The HSP increased the resistance in plants through jasmonic acid signal transduction pathway against *Puccinia striiformis* f. sp. *tritici* caused stripe rust of wheat [63]. Similarly, the expression of *OsHSP70* increased in rice plants after rice stripe virus (RSV) infection, indicating that OsHSP70 is an essential component for resistance against RSV pathogen [64]. In our study, 10 HSP genes were upregulated after *T. controversa* infection, which showed that HSP genes play a role in disease resistance against these pathogens. 

According to GO enrichment analysis, photosynthetic membrane, transferase activity, kinase activity, oxidoreductase activity, metabolic process and oxidoreductase activity has the highest numbers during plant pathogen interaction (Figure 3). These results suggested that after pathogen infection, plants activate the above pathways to counter the effect of pathogens, while KEGG enrichment analysis showed that metabolic pathway, corban metabolism, starch and sucrose metabolism and glycolysis pathways were highly activated (Figure 4), which were supported by other studies [10,65,66,67,68]. 

## 5. Conclusions

It is expected that pathogenesis-related proteins, thaumatin like proteins and heat shock were mostly identified after *T. controversa* and *T. foetida* infection, and these groups of proteins increased the resistance response in the plants against pathogens.

## Figures and Tables

**Figure 1 biology-11-00865-f001:**
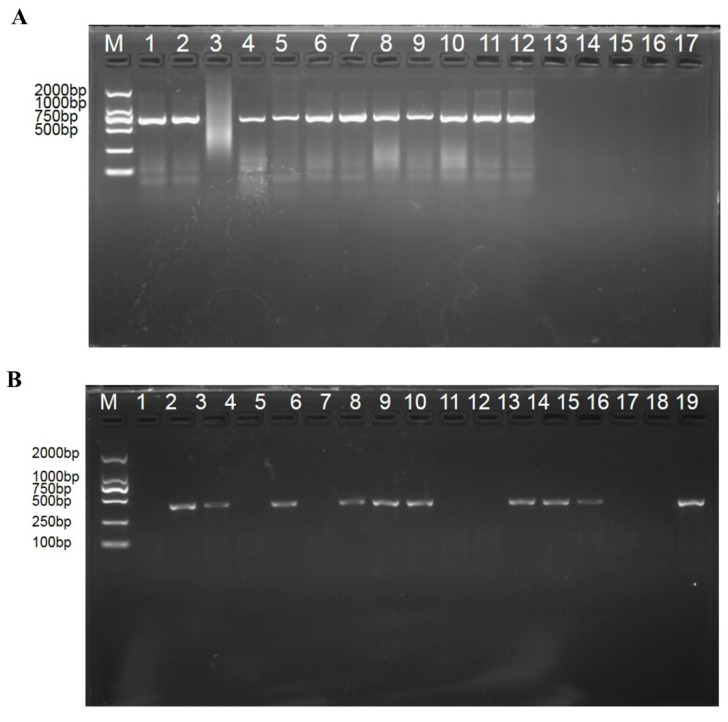
Molecular characterization of *T. controversa* and *T. foetida*. (**A**) Molecular detection of *T. foetida* from the inoculated plants. Lane 1: the positive control; lane 2–12: plants inoculated with *T. foetida*; lane 13: ddH_2_O; Lane 14–17: healthy wheat tissue. (**B**) Molecular detection of *T. controversa* from the inoculated plants. Lane 1: ddH_2_O; lane 2–18: plants inoculated with *T. controversa*; lane 19: positive control. M indicates the molecular marker DL2000.

**Figure 2 biology-11-00865-f002:**
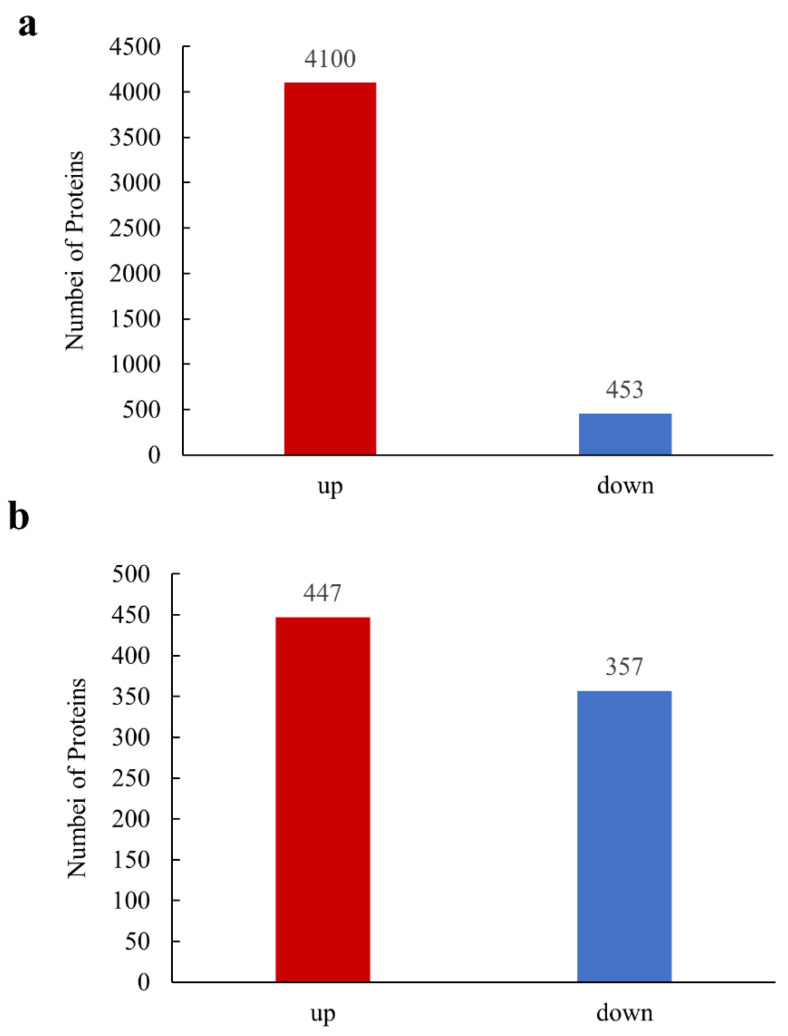
Significant differentially expressed proteins (DEPs). (**a**) Up-or downregulated DEPs in response to *T. controversa* infection; (**b**) up-or downregulated DEPs in response to *T. foetida* infection.

**Figure 3 biology-11-00865-f003:**
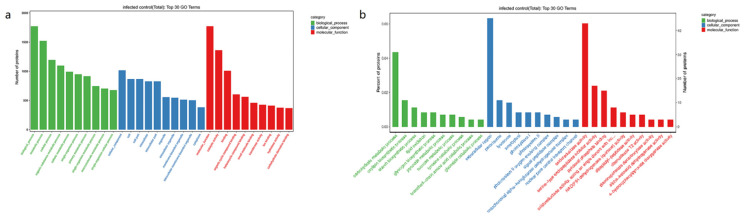
Gene ontology (GO) enrichment analysis of significant DEPs. (**a**) Gene ontology (GO) enrichment analysis of significant DEPs of *T. controversa*-infected and control libraries. (**b**) Gene ontology (GO) enrichment analysis of significant DEGs of *T. foetida*-infected and control libraries.

**Figure 4 biology-11-00865-f004:**
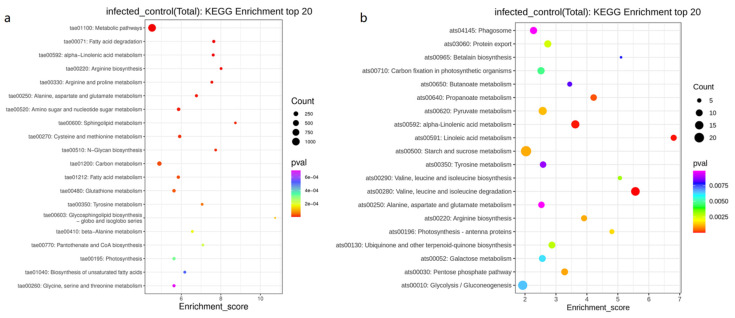
KEGG enrichment analysis scatter plot representing pathways of differentially expressed proteins. (**a**) Differentially expressed proteins of *T. controversa*-infected libraries. (**b**) Differentially expressed proteins of *T. foetida*-infected libraries.

**Table 1 biology-11-00865-t001:** Confirmation of differentially expressed proteins using parallel reaction monitoring (PRM) analysis. FC stands for “fold change”.

Protein id.	Symbol	FC (iTRAQ)	*p*-Value	FC (PRM)	*p*-Value
Reference protein	GST4	1.48	0.00003950	2.00	0.00000004
TraesCS1A02G355300.1	PR4A	2.20	0.01852322	7.48	0.00089088
TraesCS2A02G320400.1	G6PDH	1.64	2.61 × 10^−14^	1.10	0.00005540
TraesCS2D02G123300.1	SODCC.3	1.23	0.00422247	0.88	0.29967867
TraesCS4A02G106400.1	MPK5	1.76	0.00196922	2.52	0.00144951
TraesCS5A02G018200.1.cds1	RASTL-4	5.79	0.00210438	40.01	0.00000303
TraesCS5D02G364600.1	PER2	2.05	0.00019080	17.69	0.00000019
TraesCS7A02G198900.1.cds1	PRMS	2.28	0.00033301	31.00	0.00000006
TraesCS7A02G424100.1	PER22	1.34	0.00002050	1.26	0.00412683
TraesCS7D02G299500.1	MDAR5	1.23	0.00000550	1.42	0.16694490

## Data Availability

The proteomics data is available at the following link; https://www.iprox.cn/cas/login?service=https%3A%2F%2Fwww.iprox.cn%2Flogin%2Fcas, (accessed on 19 May 2022, Username: Sara, code: SARA1880016, Project ID IPX0004463000).

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
