# Peer review of "ITRAQ-Based Proteomic Analysis of Wheat (Triticum aestivum) Spikes in Response to Tilletia controversa Kühn and Tilletia foetida Kühn Infection, Causal Organisms of Dwarf Bunt and Common Bunt of Wheat"

_biology, 2022, doi:10.3390/biology11060865_

Round 1

Reviewer 1 Report

Ting et al. investigated proteomic changes of wheat plants infected with two fungal pathogens. The authors used iTRAQ analysis to identify differentially expressed proteins potentially involved in wheat antifungal responses. The identified proteins are involved in antioxidant, plant-pathogen interaction and glutathione metabolism. Of note, quantification of iTRAQ-identified proteins was confirmed by parallel reaction monitoring.

The study is interesting, describes useful resources for plant pathologists, and it is within the journal scope.

There are some concerns that should be addressed:
1. Abstract: I would suggest to clarify some aspects by replacing "4553 proteins were expressed" with "4553 proteins were differentially expressed", "804 proteins were detected" with "804 differentially expressed proteins were detected", "16 proteins after T. foetida were expressed" with "16 proteins after T. foetida were differentially expressed"
2. Materials and methods should be enhanced, for instance:
- "added protease inhibitor and extraction buffer in a certain ratio" is unclear;
- "uniprotproteome_UP000009096 databases" is unclear, please provide the complete name and the reference of the wheat database used;
- This is a well-done study since 3 replicates for each tested condition were used as indicated in supporting tables. Number of biological replicates used should be clearly stated in the main text to enhance it.
3. Figure resolution should be enhanced since some of them are hardly readable (e.g. Fig. 3)
4. L.326, it would be nice to note that peroxidase and catalase proteins were also identified in iTRAQ analysis of immune responses to plant virus infection (Pasin et al. 2014 https://doi.org/10.1371/journal.ppat.1003985 )
5. I would recommend to make raw proteomic data available in public repositories such as the Proteomics Identification Database https://www.ebi.ac.uk/pride/ or ProteomeXchange http://www.proteomexchange.org/.

Author Response

Reviewer 1

Concern 1

Abstract: I would suggest to clarify some aspects by replacing "4553 proteins were expressed" with "4553 proteins were differentially expressed", "804 proteins were detected" with "804 differentially expressed proteins were detected", "16 proteins after T. foetida were expressed" with "16 proteins after T. foetida were differentially expressed"

Response

Thank you. We replaced 4553 proteins were expressed" with "4553 proteins were differentially expressed", "804 proteins were detected" with "804 differentially expressed proteins were detected", "16 proteins after T. foetida were expressed" with "16 proteins after T. foetida were differentially expressed in the revised manuscript and highlighted with red color.

Concern 2

Materials and methods should be enhanced, for instance:

  1. i. "added protease inhibitor and extraction buffer in a certain ratio" is unclear;

Response

Thank you. We added the volume of protease inhibitor and extraction buffer in revised manuscripts. Line 137; extraction buffer in a ratio of 0.5 mol/L Tris–HCl pH 8.3, vortexed for 10 min at 4°C.

  1. ii. "uniprotproteome_UP000009096 databases" is unclear, please provide the complete name and the reference of the wheat database used;

Response

Thank you. We added the reference in the revised manuscript of the wheat database used. Reference number 34, Line 199

Zhang, R.; Jiang, W.; Liu, X.; Duan, Y.; Xiang, L.; Wang, Y.; Jiang, Y.; Shen, X.; Chen, X.; Yin, C.; et al. ITRAQ-based quantitative proteomic analysis of Fusarium moniliforme (Fusarium verticillioides) in response to Phloridzin inducers. Proteome Sci. 2021, 19, 2, doi:10.1186/s12953-021-00170-2.

Concern 3

This is a well-done study since 3 replicates for each tested condition were used as indicated in supporting tables. Number of biological replicates used should be clearly stated in the main text to enhance it.

Response

Thank you. We mentioned biological replicates in the main text in revised manuscript. Line 117-119; Three biological replicates and three technical replicates were used for fungal infected and control samples. 

Concern 4

Figure resolution should be enhanced since some of them are hardly readable (e.g. Fig. 3)

Response

Thank you. We replaced the Figure 3 in the revised manuscript with enhanced resolution.

Concern 5

L.326, it would be nice to note that peroxidase and catalase proteins were also identified in iTRAQ analysis of immune responses to plant virus infection (Pasin et al. 2014 https://doi.org/10.1371/journal.ppat.1003985 ).

Response

Thank you. We added this reference in the revised manuscript. Line 342-343; Additionally, peroxidase and catalase proteins were also identified in iTRAQ analysis of immune responses to plant virus infection [49].

Pasin, F.; Simón-Mateo, C.; García, J.A. The Hypervariable amino-terminus of P1 protease modulates potyviral replication and host defense responses. PLoS Pathog. 2014, 10, e1003985, doi:10.1371/journal.ppat.1003985.

Concern 6

I would recommend to make raw proteomic data available in public repositories such as the Proteomics Identification Database https://www.ebi.ac.uk/pride/ or ProteomeXchange http://www.proteomexchange.org/. Project ID IPX0004463000

Response

Thank you. We added the contents in line 208-210. “The raw data were available on ProteomeXchange https://www.iprox.cn/cas/login?service=https%3A%2F%2Fwww.iprox.cn%2Flogin%2Fcas, (Username Sara, code: SARA1880016, Project ID IPX0004463000)”

Reviewer 2 Report

In the manuscript, the authors have made an effort to understand the differential proteomic expression after two fungal pathogens infection of wheat. As a result, they could find several hundred differentially expressed proteins of different functionality that possibly contributed to the fungal infection. To do so, they used iTRAQ labeling and claim that this is the first documented iTRAQ labeling proteomics in wheat spikes for common and dwarf bunt fungi disease.

However, this reviewer has several concerns about the study.

  • The authors should provide a visual image of the wheat spikes infected with fungi used for the study.
  • Please provide the schematic diagram explaining the iTRaq labeling and the downstream proteomic processes to understand the study clearly.
  • The proteomic data analysis part in the material method section is poorly narrated. Please provide detailed information on how the peptides were assigned to the protein—specifically the common peptides. And how they were normalized.
  • How was the differential expression of proteins calculated? This needs to be explained clearly in detail, explaining all the criteria used.
  • In line 16, the word "expressed" is not correct. It should be detected, as it is impossible to detect all the proteins by mass spec.
  • The sentences in lines no 97- 99 are conflicting. Why did the authors wrap the plates with aluminum foil when they put them under 24 hours of light? Light cannot penetrate the aluminum foil.
  • "a certain ratio" in line 126 is ambiguous. Please correct it.
  • What is the identity of the "reference protein" in line 198?
  • In figure 1a, lanes no 15-17 are perhaps blank. Please mention it in the figure legends.
  • The labeling in figure 3a x-axis is of poor resolution and not readable, similarly in figure 4 labeling.
  • Also, the figure legends should contain detail of the figure so that readers do not have to rely on the main text to understand the figure heavily.

Author Response

Reviewer 2

Concern 1

The authors should provide a visual image of the wheat spikes infected with fungi used for the study.

Response

Thank you. We provide the figure as supplementary Figure 2.

Concern 2

Please provide the schematic diagram explaining the iTRaq labeling and the downstream proteomic processes to understand the study clearly.

Response

Thank you. We provide supplementary Figure 1 in the manuscript.

Concern 3

The proteomic data analysis part in the material method section is poorly narrated. Please provide detailed information on how the peptides were assigned to the protein—specifically the common peptides.

Response

Thank you. We improved the method section of proteomic data analysis in the revised manuscripts. Lines 199-202; PEAKS DB were searched with a fragrant ion mass tolerance of 0.05 Da and a parent ion tolerance of 10 ppm. Carbamdomethylation (C) and Itraq 8plex (K, N-term) were specified as the fixed modification. Oxidation (M), Deamidation (NQ) were specified as the variable modification.

Concern 4

And how they were normalized.

Response

Thank you. Line 204; while 1% FDR and 1 unique was used for filter the peptides for normalizing the data

Concern 5

How was the differential expression of proteins calculated? This needs to be explained clearly in detail, explaining all the criteria used.

Response

Thank you.  In line 206-208, we added “Proteins with fold change in a comparison >1.2 or <0.83 and unadjusted significance level P<0.05 were considered deferentially expressed” [36]  we also added the following reference in the reference list as its number is 36.

Ma, Q.; Shi, C.; Su, C.; Liu, Y. Complementary analyses of the transcriptome and iTRAQ proteome revealed mechanism of ethylene dependent salt response in bread wheat (Triticum aestivum L.). Food Chem. 2020, 325, 126866, doi:10.1016/j.foodchem.2020.126866.

Concern 6

In line 16, the word "expressed" is not correct. It should be detected, as it is impossible to detect all the proteins by mass spec.

Response

Thank you. We changed “expressed” to “detected” in revised manuscript. Line 23; 4553 proteins were differentially detected after T. controversa infection,

Concern 7

The sentences in lines no 97- 99 are conflicting. Why did the authors wrap the plates with aluminum foil when they put them under 24 hours of light? Light cannot penetrate the aluminum foil.

Response

Thank you. The plates were wrapped with aluminium foil for safe placing in the incubator, not for controlling the penetration of light into the plates

Concern 8

"a certain ratio" in line 126 is ambiguous. Please correct it.

Response

Thank you. We replaced certain ratio with 0.5 mol/L Tris–HCl pH 8.3 in the revised manuscript. Line 137

Concern 9

What is the identity of the "reference protein" in line 198?

Response

Thank you. We mentioned the identity of reference protein in the Table 1.

Concern 10

In figure 1a, lanes no 15-17 are perhaps blank. Please mention it in the figure legends.

Response

Thank you. Lanes 15-17 stands for healthy wheat tissues and we mentioned in revised manuscript. Line 236

Concern 11

The labeling in figure 3a x-axis is of poor resolution and not readable, similarly in figure 4 labeling.

Response

Thank you. We replaced Figure 3 and Figure 4 in the revised manuscript.

Concern 12

Also, the figure legends should contain detail of the figure so that readers do not have to rely on the main text to understand the figure heavily.

Response

Thank you. All figure legends contained the detail information of figures in the revised manuscript

Reviewer 3 Report

The manuscript is an interesting and well-prepared study aimed at identifying proteins involved in the response of plants to the attack of fungal pathogans.
however, the manuscript needs to be revised before being published.
e.g. lines 35-36 - what do DBD and CBD abbreviations mean,
the description of the maeod also requires correction, e.g. there is no citation in the primers used to identify pathogens - who developed these markers? The description of protein isolation is very unclear - what are the compositions of the buffers used for isolation?
in the discussion the authors focus on ROS, what about the other proteins identified? - they only devote a few lines to them, in my opinion the discussion should be developed. The authors obtained interesting results but did not discuss them to explain them.
Conclusions: in my opinion, the authors wrote only a laconic summary, not conclusions. in this part of the work, they should indicate which groups of proteins they have identified and how these proteins may influence the plant's response to pathogen attack.

Author Response

Reviewer 3

Concern 1

e.g. lines 35-36 - what do DBD and CBD abbreviations mean,

Response

Thank you. We changed DBD to DBW (dwarf bunt of wheat) and CBD to CBW (common bunt of wheat) in revised manuscript. Lines 43-44

Concern 2

the description of the methods also requires correction, e.g. there is no citation in the primers used to identify pathogens - who developed these markers?

Response

Thank you.  We added the references who developed these markers in the revised manuscript. The reference number 32 and 33. Line 130 and 132

Gao, L.; Yu, H.; Han, W.; Gao, F.; Liu, T.; Liu, B.; Kang, X.; Gao, J.; Chen, W. Development of a SCAR marker for molecular detection and diagnosis of Tilletia controversa Kühn, the causal fungus of wheat dwarf bunt. World J. Microbiol. Biotechnol. 2014, 30, 3185–3195.

Yao, Z.; Qin, D.; Chen, D.; Liu, C.; Chen, W.; Liu, T.; Liu, B.; Gao, L. Development of ISSR-derived SCAR marker and SYBR Green I real-time PCR method for detection of teliospores of Tilletia laevis Kühn. Sci. Rep. 2019, 9, 17651, doi:10.1038/s41598-019-54163-5.

Concern 3

The description of protein isolation is very unclear - what are the compositions of the buffers used for isolation?

Response

Thank you. We added the compositions of the buffers used for isolation in the revised manuscripts. Line 137; extraction buffer in a ratio of 0.5 mol/L Tris–HCl pH 8.3, vortexed for 10 min at 4°C.

Concern 4

in the discussion the authors focus on ROS, what about the other proteins identified? - they only devote a few lines to them, in my opinion the discussion should be developed. The authors obtained interesting results but did not discuss them to explain them.

Response

Thank you. We added possible discussion in the revised manuscript with suitable references. Lines 342-344; Additionally, peroxidase and catalase proteins were also identified in iTRAQ analysis of immune responses to plant virus infection [49].

Pasin, F.; Simón-Mateo, C.; García, J.A. The Hypervariable amino-terminus of P1 protease modulates potyviral replication and host defense responses. PLoS Pathog. 2014, 10, e1003985, doi:10.1371/journal.ppat.1003985

Lines 347-349; Additionally, the catalase enzymes have antioxidant action and can prevent the formation of free radicals, sequester them or even degrade them, thus minimizing or preventing damage to plant cells [52].

Pittner, E.; Marek, J.; Bortuli, D.; Santos, L.A.; Knob, A.; Faria, C.M.D.R. Defense responses of wheat plants (Triticum aestivum L.) against brown spot as a result of possible elicitors application. Arq. Inst. Biol. (Sao. Paulo). 2019, 86, 1–16, doi:10.1590/1808-1657000312017

Lines 360-364; Pathogenesis related proteins (PRs) have key roles in defense mechanism of the wheat crops against T. controversa. PRs are a group of functionally diverse inducible proteins that accumulate in plant tissue in response to fungal infection [58]. The expression profiles of PRs significantly high in T. controversa infected plants than control plants [9].

van Loon, L.C.; Rep, M.; Pieterse, C.M.J. Significance of inducible defense-related proteins in infected plants. Annu. Rev. Phytopathol. 2006, 44, 135–162.

Muhae-Ud-Din, G.; Chen, D.; Liu, T.; Chen, W.; Gao, L. Characterization of the wheat cultivars against Tilletia controversa Kühn, causal agent of wheat dwarf bunt. Sci. Rep. 2020, 10, 9029, doi:10.1038/s41598-020-65748-w.

Lines 367-271; Similarly, glutathione play key roles within different cell compartments, in the development of resistance and activating the plant defense. The accumulation of glutathione in peroxisomes and chloroplasts at the early stages of plant pathogen interactions is related to increased resistance and tolerance against different pathogens [62].

Zechmann, B. Subcellular roles of glutathione in mediating plant defense during biotic stress. Plants 2020, 9, 1067, doi:10.3390/plants9091067.

Concern 5

Conclusions: in my opinion, the authors wrote only a laconic summary, not conclusions. in this part of the work, they should indicate which groups of proteins they have identified and how these proteins may influence the plant's response to pathogen attack

Response

Thank you. We changed the conclusion in the revised manuscript. Line 391-393; It is expected that pathogenesis-related proteins, thaumatin like proteins and heat shock were mostly identified after T. controversa and T. foetida infection and these groups of proteins increased the resistant response in the plants against pathogens.

Round 2

Reviewer 3 Report

the authors corrected the manuscript in line with the comments. the work can be published